# Long-Term Immunogenicity and Safety of a Homologous Third Dose Booster Vaccination with TURKOVAC: Phase 2 Clinical Study Findings with 32-Week Post-Booster Follow-Up

**DOI:** 10.3390/vaccines12020140

**Published:** 2024-01-29

**Authors:** Zafer Sezer, Shaikh Terkis Islam Pavel, Ahmet Inal, Hazel Yetiskin, Busra Kaplan, Muhammet Ali Uygut, Ahmet Furkan Aslan, Adnan Bayram, Mumtaz Mazicioglu, Gamze Kalin Unuvar, Zeynep Ture Yuce, Gunsu Aydin, Refika Kamuran Kaya, Ihsan Ates, Ates Kara, Aykut Ozdarendeli

**Affiliations:** 1Department of Medical Pharmacology, Faculty of Medicine, Erciyes University, Kayseri 38280, Türkiye; zsezer@erciyes.edu.tr (Z.S.); ainal@erciyes.edu.tr (A.I.); 2Good Clinical Practise Centre (IKUM), Erciyes University, Kayseri 38280, Türkiye; 3Vaccine Research, Development and Application Centre (ERAGEM), Erciyes University, Kayseri 38280, Türkiye; pavel@erciyes.edu.tr (S.T.I.P.); hazelyetiskin@erciyes.edu.tr (H.Y.); busrakaplan@erciyes.edu.tr (B.K.); mauygut@erciyes.edu.tr (M.A.U.); afurkanaslan@erciyes.edu.tr (A.F.A.); gunsuaydin@erciyes.edu.tr (G.A.); 4Department of Anesthesiology and Reanimation, Faculty of Medicine, Erciyes University, Kayseri 38280, Türkiye; adnanbayram@erciyes.edu.tr; 5Department of Family Medicine, Faculty of Medicine, Erciyes University, Kayseri 38280, Türkiye; mazici@erciyes.edu.tr; 6Department of Infectious Diseases and Clinical Microbiology, Faculty of Medicine, Erciyes University, Kayseri 38280, Türkiye; gamzekalinunuvar@erciyes.edu.tr (G.K.U.); zeynepture@erciyes.edu.tr (Z.T.Y.); 7Department of Microbiology, Faculty of Medicine, Erciyes University, Kayseri 38280, Türkiye; 8Health Institutes of Türkiye (TUSEB), Istanbul 34718, Türkiye; kamuran.kaya@tuseb.gov.tr (R.K.K.); ateskara@hacettepe.edu.tr (A.K.); 9Department of Internal Medicine, University of Health Sciences Ankara City Hospital, Ankara 06530, Türkiye; ihsan.ates@saglik.gov.tr; 10Department of Pediatrics, Pediatric Infectious Disease, Faculty of Medicine, Hacettepe University, Ankara 06430, Türkiye

**Keywords:** booster, COVID-19, immunogenicity, inactivated vaccine, neutralizing antibody, S1 RBD, safety, SARS-CoV-2, seroconversion, TURKOVAC

## Abstract

Vaccine-induced immunity wanes over time and warrants booster doses. We investigated the long-term (32 weeks) immunogenicity and safety of a third, homologous, open-label booster dose of TURKOVAC, administered 12 weeks after completion of the primary series in a randomized, controlled, double-blind, phase 2 study. Forty-two participants included in the analysis were evaluated for neutralizing antibodies (NAbs) (with microneutralization (MNT_50_) and focus reduction (FRNT_50_) tests), SARS-CoV-2 S1 RBD (Spike S1 Receptor Binding Domain), and whole SARS-CoV-2 (with ELISA) IgGs on the day of booster injection and at weeks 1, 2, 4, 8, 16, 24, and 32 thereafter. Antibody titers increased significantly from week 1 and remained higher than the pre-booster titers until at least week 4 (week 8 for whole SARS-CoV-2) (*p* < 0.05 for all). Seroconversion (titers ≥ 4-fold compared with pre-immune status) persisted 16 weeks (MNT_50_: 6-fold; FRNT_50_: 5.4-fold) for NAbs and 32 weeks for S1 RBD (7.9-fold) and whole SARS-CoV-2 (9.4-fold) IgGs. Nine participants (20.9%) tested positive for SARS-CoV-2 RT-PCR between weeks 8 and 32 of booster vaccination; none of them were hospitalized or died. These findings suggest that boosting with TURKOVAC can provide effective protection against COVID-19 for at least 8 weeks and reduce the severity of the disease.

## 1. Introduction

Coronavirus disease 2019 (COVID-19) is a highly contagious disease caused by severe acute respiratory syndrome coronavirus 2 (SARS-CoV-2), which became a pandemic in March 2020, three months after China reported the first case [1]. As of 31 August 2023, the infection had affected more than 770 million people worldwide and caused approximately 7 million deaths [2].

The high transmissibility of SARS-CoV-2 has made vaccination a key pillar of the fight against COVID-19 [3]. Tremendous efforts have been made to develop, manufacture, and distribute safe and effective vaccines against SARS-CoV-2 to reduce the spread and severity of the infection and the associated hospitalizations and deaths [3,4,5,6].

Knowledge gained about family *Coronaviridae* during severe acute respiratory syndrome (SARS) and Middle East respiratory syndrome (MERS) outbreaks, advances in vaccine technology, and collaboration between academia, manufacturers, regulatory agencies, and funding organizations have enabled an accelerated COVID-19 vaccine development process without compromising safety and quality [3,4,7]. Several vaccines became available outside a clinical trial setting within a year after the infection first appeared [4]. As of July 2023, there have been 13 COVID-19 vaccines authorized for emergency use by the World Health Organization (WHO), and hundreds of vaccine candidates are in various stages of development [8].

Türkiye was one of the first countries to initiate research on COVID-19 vaccine development [9]. ERUCoV-VAC, later named TURKOVAC, is an inactivated whole-virion SARS-CoV-2 vaccine developed under the national vaccine development program. Preclinical and interim phase 1 (NCT04691947) and 2 (NCT04824391) trial results of the vaccine have been previously published [10,11]. Based on the immunogenicity and safety findings from these trials, a regimen of two intramuscular (im) injections of TURKOVAC 3 µg administered 28 days apart is recommended for primary immunization [11]. The vaccine has been available in Türkiye since December 2021 with emergency use authorization granted by the Turkish Ministry of Health, and the development program is ongoing [12].

Although COVID-19 is no longer considered a Public Health Emergency of International Concern (PHEIC) [13] as of May 2023, it remains an ongoing health issue due to the emergence of new variants and the waning vaccine-induced immune responses over time. Therefore, booster vaccination has been suggested, especially for at-risk populations, to enhance immunity against SARS-CoV-2 [14].

The Hybrid COV-RAPEL TR Study (NCT04979949) demonstrated that heterologous boosting with TURKOVAC 90 to 270 days after receiving two doses of the CoronaVac vaccine stimulated a significant immune response that persisted up to post-booster Day 84 with acceptable safety and tolerability [15]. However, there was a gap in knowledge about the outcomes of homologous boosting with TURKOVAC. Therefore, we investigated the long-term (32 weeks) immunogenicity, safety, and efficacy of a third, homologous, open-label booster dose of the vaccine in healthy adults administered 12 weeks after completion of the primary series in a randomized, placebo-controlled, double-blind, phase 2 study.

## 2. Materials and Methods

### 2.1. Study Design and Participants

In a randomized, double-blind, placebo-controlled, phase 2 immunogenicity and safety trial of the inactivated COVID-19 vaccine TURKOVAC, healthy volunteers <65 years of age were randomly assigned (2:2:1) to receive two intramuscular injections of TURKOVAC 3 µg or 6 µg or a placebo (0.9% saline) 28 days apart. Considering the immunogenicity and safety results for the primary series [11], TURKOVAC 3 µg was selected as the optimal dose to continue the clinical development program, and the study protocol was amended to investigate the immunogenicity and safety of a booster dose of TURKOVAC 3 µg. Subjects who had received two doses of TURKOVAC 3 μg for primary immunization during the study were invited to participate in the booster substudy. Those who gave their consent to receive the booster dose and had had a recent negative reverse transcriptase polymerase chain reaction (RT PCR) test for SARS-CoV-2 received a third dose of the vaccine 12 weeks after the second dose and were followed up to 32 weeks after the booster injection.

This study was conducted according to the guidelines of the Declaration of Helsinki and approved by the Ethics Committee for Clinical Trials of Erciyes University (17 June 2021; 2021/396) and the Turkish Ministry of Health (18 June 2021; E-66175679-514.02.01-463635). Written informed consent was obtained from all subjects involved in the study. The trial is registered on ClinicalTrials.gov (NCT04824391) (10 February 2021).

### 2.2. Procedures and Outcomes

A microneutralization test (MNT_50_) and focus reduction neutralization test (FRNT_50_) were performed to measure neutralizing antibodies (NAbs) to wild-type SARS-CoV-2 (hCoV-19/Türkiye/ERAGEM-001/2020 strain, GenBank accession number; MT327745.1 and GISAID; EPI_ISL_424366). IgG responses to SARS-CoV-2 S1 RBD (Spike S1 Receptor Binding Domain) and whole SARS-CoV-2 were evaluated with the Euroimmune anti-SARS CoV-2 IgG enzyme linked immunoassay (ELISA) kit and in-house IgG ELISA (based on purified whole SARS-CoV-2), respectively. Methods of immunogenicity testing were previously reported in detail [11,16,17]. Laboratory investigations for immunogenicity were performed on the day of booster injection (i.e., second dose +12 weeks) and at weeks 1, 2, 4, 8, 16, 24, and 32 thereafter. The geometric mean titers (GMTs) of the antibodies were compared to the pre-booster (second dose +12 weeks) levels. A ≥4-fold higher post-booster antibody titer compared to the pre-immune levels served as an immune correlate of protection (ICP) predicting the clinical efficacy of the booster dose.

Adverse event (AE) questioning and laboratory (blood chemistry and hematology) investigations for safety were performed on the same days as the immunogenicity assessments. In addition, daily phone calls were made to collect AEs within the first week of booster injection. AEs were graded as mild (grade 1: requiring no intervention; no impact on activities of daily living (ADL)), moderate (grade 2: requiring minimal, non-invasive intervention; moderate impact on ADL); and severe (grade 3: requiring invasive intervention; major assistance needed for ADL).

### 2.3. Statistical Analysis

The GraphPad Prism 9.0.1 program was used for statistical analyses and graphical representations of immunogenicity data. Antibody titers were presented as GMTs including 95% confidence interval (CI) and seroconversion rates (number of patients and %). An unpaired t-test was used to compare the antibody titers; Spearman’s correlation curves and linear regression analyses were utilized to assess the correlation between MNT_50_ and FRNT_50_ results at pre-determined study time-points.

All volunteers who received a booster dose of TURKOVAC 3 µg constituted the safety population. AEs were descriptively analyzed as number and percentage of events.

A *p* value < 0.05 was considered statistically significant for all tests.

## 3. Results

Out of 93 study participants who had received two doses of TURKOVAC 3 µg 28 days apart for primary immunization, 43 (46.2%) agreed to receive a booster dose of the vaccine. The mean age of these subjects was 36.79 ± 10.20 years (range: 20–57), and 33 of them (76.7%) were men. Their mean body mass index was 25.7 ± 3.7 (range: 18.3–32.0). Forty-two patients were eligible and included in the analysis.

### 3.1. Immunogenicity

Table 1 presents the GMTs of NAbs, anti-S1-RBD, and anti-whole SARS-CoV-2 IgG antibodies and the seroconversion rates at baseline (pre-immune) on the day of booster injection (12 weeks after the second dose of primary series; pre-booster) and at weeks 1, 2, 4, 8, 16, 24, and 32 thereafter. The changes in antibody titers over the course of follow-up are shown in Figure 1, including how many times GMTs increased at each time-point compared to pre-immune levels.

At 12 weeks after the second vaccination, before the booster shot, NAb seroconversion persisted in approximately 80% of subjects (Table 1), with 5.2-fold and 4.5-fold higher NAb GMTs in MNT_50_ and FRNT_50_ assays compared to the pre-immune levels, respectively (Figure 1A,B). Significant increases in NAb GMTs occurred from 1 week after booster vaccination compared to pre-booster levels, peaking at week 2 and persisting until week 4 (*p* < 0.05 for all). The NAb titers then showed a gradual decline and became comparable to pre-booster levels at weeks 8, 16, 24, and 32. However, they remained ≥4-fold higher than at pre-immune status at weeks 8 and 16 after the booster shot. The seroconversion rates for NAbs were below 50% and their GMTs were four times lower than the pre-immune levels at weeks 24 and 32 (Table 1 and Figure 1A,B). The results of MNT_50_ and FRNT_50_ assays were very strongly correlated at all assessment time-points and showed a perfect correlation at week 32 (r = 1; *p* = 0.001) (Figure 2).

As presented in Table 1, the seroconversion rates for anti-S1-RBD and anti-SARS-CoV-2 IgG antibodies on the day of booster injection were 100% and 92.8%, respectively. The GMTs of both antibodies significantly increased, and all subjects achieved seroconversion at week 1 after the booster injection (*p* < 0.0001 for both) with 63.5-fold and 48.7-fold-higher GMTs for anti-S1-RBD and anti-whole SARS-CoV-2 IgGs compared to the pre-immune values, respectively. IgG antibody titers peaked 2 weeks after the booster shot and gradually declined in subsequent visits. The anti-S1-RBD IgG GMTs at weeks 8, 16, 24, and 32 after the third injection were comparable to the pre-booster level but remained ≥4 fold higher than the titer at the pre-immune state at all these time-points. The anti-whole SARS-CoV-2 IgG GMT also peaked 2 weeks after the booster dose administration. Unlike the anti-S1-RBD IgG, the GMTs of anti-whole SARS-CoV-2 IgG at weeks 8 and 16 were significantly higher than the pre-booster level (*p* < 0.0001 and *p* < 0.005, respectively). At week 32 after the booster dose, the GMTs of anti-S1-RBD and anti-whole SARS-CoV-2 IgG antibodies were 7.9-fold and 9.4-fold higher than the pre-immune levels, respectively (Figure 1C,D). The percentage of seroconverted patients was 76.9% for anti-S1-RBD IgG and 61.5% for the anti-whole SARS-CoV-2 IgG at this time-point.

### 3.2. Safety

Table 2 provides a summary of the 46 AEs experienced during the post-booster 32 weeks. None of these events were severe. Almost two thirds of the events (63.3%; 30 AEs in 19 participants) occurred after the 8th week of booster shot. Headache (*n* = 7; 43.8%) was the most common AE experienced within the initial 8 weeks that followed the booster injection.

None of the study participants had a laboratory-confirmed SARS-CoV-2 infection within 8 weeks of booster injection. In total, nine participants were tested positive for COVID-19 by SARS-CoV-2 RT-PCR after the eight week of booster vaccination. Among these cases, three were diagnosed between weeks 8 and 16, with neutralization titers ranging from 1/8 to 1/16. The remaining six cases were detected beyond week 16, also with neutralization titers ranging from 1/8 to 1/16, except for one case which had a negative neutralization titer. None of the infected patients had a severe disease requiring hospital admission. There were no deaths associated with COVID-19.

Eleven subjects (26.2%) had an overall 15 abnormal laboratory test results requiring repeat testing within the same period; abnormal blood glucose levels (*n* = 9) in six subjects (14.3%) were the most common laboratory abnormalities, followed by abnormal white blood cell counts (*n* = 3) in three subjects (7.1%) and abnormal blood urea nitrogen levels in two subjects (4.8%).

## 4. Discussion

We found that a homologous booster shot with TURKOVAC, administered 12 weeks after the completion of primary immunization against SARS-CoV-2, elicited rapid and robust immune responses with acceptable safety and tolerability in healthy adults <65 years of age. Overall, the results of this study are consistent with those of the previously published studies that investigated the immunogenicity and safety of homologous boosting with inactivated vaccines against SARS-CoV-2 [18,19,20,21,22,23,24,25,26].

Previous studies on inactivated COVID-19 vaccines have demonstrated that the humoral immune responses elicited by a two-dose primary immunization gradually diminished over time, typically remaining detectable for up to 6 months following the second dose [15,18,19,20,21,22,23,24,25,26,27,28]. Ates et al. conducted an investigation to assess the long-term immunogenicity of the TURKOVAC and CoronaVac vaccines when administered as booster doses subsequent to the second dose of primary vaccination with CoronaVac. Their findings revealed a slight decline in antibody positivity on Day 84 compared to Day 28; however, there was no statistically significant difference observed between the two vaccine groups in terms of antibody response [18]. The study conducted by Zeng et al. investigated the immune persistence and efficacy of CoronaVac, a two-dose COVID-19 vaccine, in individuals aged 18 years and older. The results indicate that after a period of six months, the levels of neutralizing antibodies induced by the two-dose regimen of CoronaVac declined to low concentrations. However, the administration of a third dose, eight months after the second dose, led to a significant enhancement in the immune response, with neutralizing antibody levels increasing three-fold to five-fold. This study also demonstrated the safety of the third dose, as no adverse events were reported, and the reactogenicity of the vaccine was comparable to that of the placebo. Notably, regardless of age group, a high seropositivity rate ranging from 98% to 100% was achieved after the administration of the third dose. These findings suggest that the third dose of CoronaVac, given at an interval of eight months after the second dose, substantially augments neutralizing antibody levels, potentially conferring longer-lasting immunity and a heightened level of protection compared to the standard two-dose schedule [27]. AI et al. conducted a study to evaluate the immunogenicity and safety of a third homologous BBIBP-CorV booster vaccination administered four to eight months after the initial two doses. The results demonstrated that the third dose of BBIBP-CorV was well tolerated and highly immunogenic in healthy adults aged 18–59 years. This study presented additional evidence demonstrating the effectiveness of a third dose in generating strong humoral and cell-mediated immune responses, specifically targeting variants of concern (VOCs). The administration of a third dose of BBIBP-CorV vaccine effectively stimulated and promptly elevated the humoral immune response by enhancing antibody levels. Moreover, the third dose demonstrated both safety and efficacy in eliciting robust humoral and cell-mediated immune responses. These findings provide support for the potential adoption of a third homologous BBIBP-CorV booster vaccination approach to enhancing and extending protection against COVID-19 [28].

The administration of a third dose, utilizing different vaccine platforms in addition to inactivated vaccines, has been shown to rapidly enhance the immune response and maintain its effectiveness for an extended period. The safety and immunogenicity evaluation of a booster dose of the BNT162b2 vaccine, given 7 to 9 months after the initial two-dose series, indicates that a third dose has the potential to extend the duration of protection and further strengthen the breadth of defense against COVID-19. These findings emphasize the scientific rationale and importance of administering a third dose to optimize and sustain immune protection, especially in the face of emerging variants and the ongoing need for long-lasting immunity in the fight against the COVID-19 pandemic [29]. Flaxman et al. investigated the immune responses to ChAdOx1 nCoV-19 following a second dose with an extended interval between the first and second dose, as well as after a third dose with an extended interval between the second and third doses. Notably, they found that prolonging the interval between the first two doses to 44–45 weeks resulted in higher antibody titers after the second dose compared to a shortened interval. Moreover, administering a third dose 28–38 weeks after the primary series led to antibody titers surpassing those observed after a second dose with a shortened interval. Importantly, the reactogenicity was lower after the second or third dose compared to the first dose [30].

In our study, the GMTs of NAbs and ELISA-detected SARS CoV-2-specific IgGs were above the seropositivity thresholds for the relevant assays on the day of booster administration, i.e., 12 weeks after completing the primary series, and the seroconversion rates were approximately 80% for NAbs and exceeded 90% for IgGs. Although these findings suggest that a substantial group of participants might have had the potential to remain seropositive for longer periods of time after primary immunization, we do not know what the impact of delaying the booster administration would be as we only tested the 12-week boosting schedule.

Protection against SARS-CoV-2 infection and a reduction in disease severity in affected individuals are complex processes in which both the humoral and cellular components are involved [25,31,32,33,34,35,36,37,38]. Various humoral markers, including anti-spike protein/anti-RBD IgG and IgA and NAbs, have been suggested as potential surrogate markers of SARS-CoV-2 vaccine efficacy, but there are no established protective thresholds or ranges for these antibodies [34,35,36,37,38]. In this study, a booster dose of TURKOVAC increased the seroconversion rate of NAbs to >95% and those of anti-SARS-CoV-2 S1 RBD IgG to 100% as early as 1 week after the injection, and >90% of the subjects remained seropositive for both antibodies for at least 8 weeks after the vaccination. None of the participants had a PCR-confirmed SARS-CoV-2 infection during this period. It is noteworthy to mention that two thirds of the confirmed cases of infection occurred after the sixteenth week of booster administration, when the GMTs of NAbs fell below six times the pre-immune levels and there were no hospitalizations or deaths due to COVID-19 throughout the 32-week study period despite the declining antibody GMTs over time. Although this study was not designed to determine an ICP, our findings suggest that the NAbs may be a potential correlate of protection at least against laboratory-confirmed SARS-CoV-2 infection for TURKOVAC. The GMTs of IgGs, which remained above the lower limit of seroconversion throughout the study period, might be explained by the persistence of specific immune memory cells allowing for antibody production following exposure to the relevant antigens. Overall, our findings show the clinical efficacy of boosting with TURKOVAC in preventing SARS-CoV-2 infection and reducing COVID-19 severity and are complementary to those from previous studies of various inactivated vaccines which reported low rates of infection, pneumonia, hospitalization, and death associated with SARS-CoV-2 infection after the administration of a booster dose [39,40,41,42,43,44,45,46].

The current study did not reveal any new concerns regarding TURKOVAC safety. All AEs were mild to moderate in severity and resolved within a few days. In contrast to other inactivated COVID-19 vaccine studies [19,20,21,23,28,29,39], including those of TURKOVAC [11,15,18,32], none of the participants in this study reported pain at the injection site after receiving a booster injection. This may be because we collected AEs through spontaneous reporting, unlike previously published TURKOVAC studies where safety assessments included both solicited and unsolicited data collection and pain at injection site was the most reported local reaction.

To our knowledge, this is the first paper to report the outcomes in volunteers who were boosted with homologous TURKOVAC vaccine. The strengths of this study are the long follow-up period extending up to 32 weeks after the booster dose and the assessment of immunogenicity with both NAbs and SARS-specific IgGs. This provides valuable information about the long-term immunogenicity and efficacy of a booster dose of TURKOVAC.

The following limitations should be considered when interpreting the results. This was a small-sized, single-arm study which included healthy adults aged <65 years and investigated the immunogenicity and safety of a single boosting scheme. In addition, this study only evaluated the antibody responses against wild-type SARS-CoV-2 and did not include cellular immune response assessments.

One of the limitations of our study is that we lack information about the specific variants or lineages with which the nine volunteers were infected, despite their positive rt-PCR results during the study. However, it is worth noting that a study conducted in Türkiye between April 2021 and February 2022 analyzed 492 SARS-CoV-2 strains. Out of these, 64% were identified as variants, while 16% were classified as the wild type. During this period, seven different lineages and a sublineage were reported among the variant sequences. Initially, the Alpha variant was dominant, followed by the Beta, Delta, Eta, and Lota variants. However, by September 2021, the Delta variant became the dominant variant in Türkiye. In December 2021, the Omicron variant was reported for the first time, and by February 2022 it overtook the Delta variant [47]. According to these results, it can be speculated that the Alpha variant was initially dominant during the study, followed by the Delta variant, and in the final stages of the study, the Omicron variant was detected for the first time.

Ongoing studies are actively investigating the vaccine’s efficacy against variants of concern (VoCs) and evaluating cellular immune responses, with these studies currently in the process of being prepared for submission.

## 5. Conclusions

The administration of a third homologous booster dose of TURKOVAC, an inactivated whole-virion SARS-CoV-2 vaccine, 12 weeks after the completion of primary immunization can safely provide effective protection against SARS-CoV-2 infection and reduce the severity of COVID-19 by inducing strong humoral immune responses which persist at least 8 weeks in healthy adults under 65 years of age. Future research and real-life data on the immunogenicity, efficacy, or effectiveness of various boosting regimens against the variants of concern in study populations, including those who are vulnerable to SARS-CoV-2 infection, will help optimize the immunization strategy for TURKOVAC.

## 6. Patents

Aykut Ozdarendeli, Shaikh Terkis Islam Pavel, Hazel Yetiskin, Muhammet Ali Uygut, and Gunsu Aydin are the named inventors on patent applications covering inactivated COVID-19 vaccine development.

## Figures and Tables

**Figure 1 vaccines-12-00140-f001:**
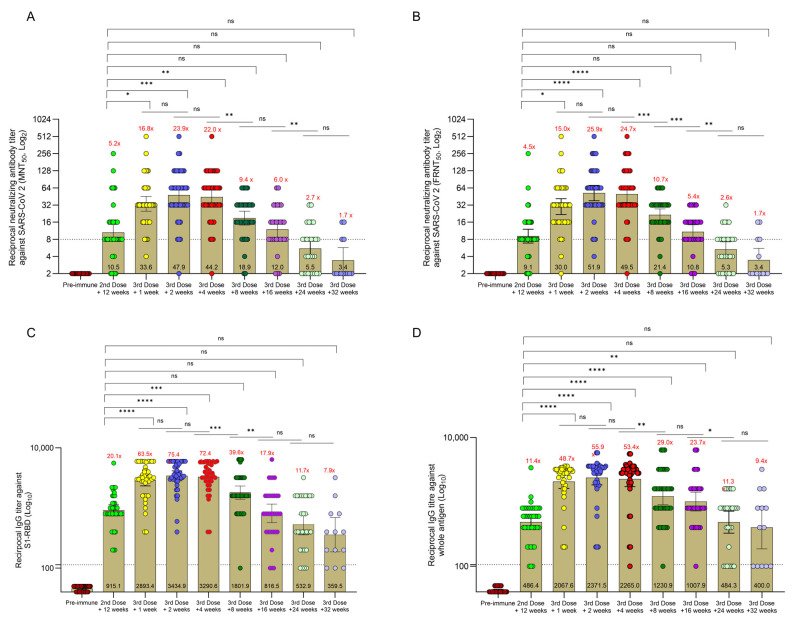
Comparison of antibody titers across the study assessment time-points. (**A**) shows the neutralizing antibody titer in the MNT_50_ (micro-neutralization test) assay. (**B**) shows the neutralizing antibody titer in the FRNT_50_ (focus reduction neutralization test) assay. (**C**) shows the IgG titer against S1-RBD. (**D**) shows the IgG titer against the whole SARS-CoV 2 antigen. The values inside the bars represent geometric mean titers (GMTs), and the values above the bars (shown in red) show how many times GMT values increased versus the pre-immune levels. The dotted line represents the threshold value for the experiments. The unpaired t-test was used to determine the statistically significant differences between groups. *p* < 0.05 indicates statistically significant differences, with ns indicating nonsignificant; * < 0.05, ** < 0.005, *** < 0.0005 and **** <0.0001.

**Figure 2 vaccines-12-00140-f002:**
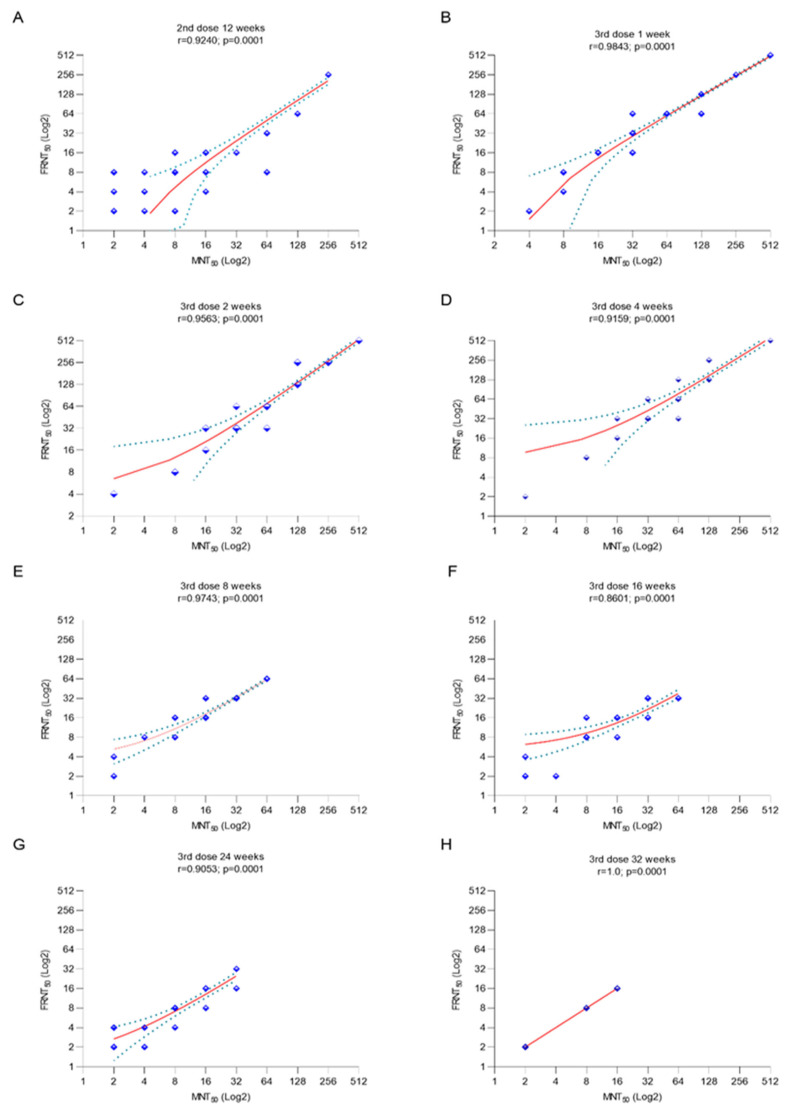
Correlation between MNT_50_ and FRNT_50_ results. (**B**–**H**). Correlation between MNT_50_ and FRNT_50_ at post-booster 1, 2, 4, 8, 16, 24, and 32 weeks. r: correlation coefficient *p* < 0.05 indicates statistical significance. ▬▬ Regression line **…….** Error bar. (**A**). Correlation between MNT_50_ and FRNT_50_ at 2nd dose, 12 weeks.

**Table 1 vaccines-12-00140-t001:** Pre-immune, pre-booster, and post-booster assessments of antibody titers and seroconversion rates. * Data are % (n/N) [95 %CI]. Seroconversion was defined as fourfold rise over baseline; n = number of participants who achieved seroconversion. N = number of participants included in the immunogenicity analysis; CI = confidence interval.

Antibody Responses	Pre-Immune	2nd Dose + 12 Weeks	3rd Dose + 1 Week	3rd Dose + 2 Weeks	3rd Dose + 4 Weeks	3rd Dose + 8 Weeks	3rd Dose + 16 Weeks	3rd Dose + 24 Weeks	3rd Dose + 32 Weeks
SARS-CoV 2-neutralizing antibodies (MNT_50_) (GMT-95%CI)	2.0 (2.0–2.0)	10.5 (3.5–23.7)	33.6 (6.7–60.4)	47.9 (21.2–74.5)	44.2 (18.9–69.4)	18.9 (12.7–25.0)	12.0 (5.6–18.3)	5.5 (2.1–8.8)	3.4 (0.1–6.6)
Seroconversion (%) * Seroconverted/tested (*n*) 95%-CI	0.0% 0/43 0.0–0.0	78.5% 33/42 63.1–89.7	97.6% 41/42 87.4–99.9	97.6% 41/42 87.4–99.9	97.6% 41/42 87.4–99.9	91.8% 34/37 78.0–98.3	85.7% 24/28 67.3–95.9	48.1% 13/27 28.6–68.0	30.7% 4/13 9.0–61.4
SARS-CoV 2-neutralizing antibodies (FRNT_50_) (GMT-95%CI)	2.0 (2.0–2.0)	9.1 (3.11–21.3)	30.0 (3.4–56.5)	51.9 (22.8–80.9)	49.5 (20.2–78.7)	21.4 (15.5–27.2)	10.8 (7.4–14.4)	5.3 (2.6–7.9)	3.4 (0.1–6.6)
Seroconversion (%) * Seroconverted/tested (*n*) 95%-CI	0.0% 0/43 0.0–0.0	80.9% 34/42 65.8–91.4	95.2% 40/42 83.8–99.4	97.6% 41/42 87.4–99.9	97.6% 41/42 87.4–99.9	94.5% 35/37 81.8–99.3	85.7% 24/28 67.3–95.9	44.4% 12/27 25.4–64.6	30.7% 4/13 9.0–61.4
Antibody responses to S1-RBD (GMT-95%CI)	45.2 (44.0–47.0)	915.1 (635.5–1194.3)	2893.4 (2347.5–3439.2)	3434.9 (2912.3–3957.4)	3290.6 (2781.8–3799.3)	1801.9 (1245.9–2357.8)	816.5 (336.6–1266.3)	532.9 (243.9–821.7)	359.5 (76.9–887.2)
Seroconversion (%) * Seroconverted/tested (*n*) 95%-CI	0.0% 0/43 0.0–0.0	100% 42/42 91.5–100.0	100% 42/42 91.5–100.0	100% 42/42 91.5–100.0	100% 42/42 91.5–100.0	97.2% 36/37 85.8–99.9	92.8% 26/28 76.5–99.1	85.1% 23/27 66.2–96.1	76.9% 10/13 46.1–94.9
Antibody responses to whole SARS-CoV-2 antigen (GMT-95%CI)	42.6 (41.2–43.9)	486.4 (326.8–645.8)	2067.6 (1754.8–2380.3)	2371.5 (1986.1–2756.9)	2265.0 (1889.0–2641.0)	1230.9 (674.2–1787.5)	1007.9 (406.1–1609.6)	484.3 (272.1–696.4)	400.0 (32.0–946.1)
Seroconversion (%) * Seroconverted/tested (*n*) 95%-CI	0.0% 0/43 0.0–0.0	92.8% 39/42 80.5–98.5	100% 42/42 91.5–100.0	100% 42/42 91.5–100.0	97.6% 41/42 87.4–99.9	97.2% 36/37 85.8–99.9	96.4% 27/28 81.6–99.9	74.0% 20/27 53.7–88.8	61.5% 8/13 31.5–86.1

**Table 2 vaccines-12-00140-t002:** Adverse events experienced after the booster injection.

Type of Event	Time from the Booster Dose		Overall *n* (%)
	0–8 Weeks *n* (%)	9–32 Weeks *n* (%)	
Positive SARS-CoV-2 RT PCR test	-	9 (30)	9 (19.6)
Headache	7 (43.8)	2 (6.7)	9 (19.6)
Weakness	1 (6.3)	4 (13.3)	5 (10.9)
Runny nose	1 (6.3)	3 (10)	4 (8.7)
Joint pain	1 (6.3)	3 (10)	4 (8.7)
Sore throat	1 (6.3)	2 (6.7)	3 (6.5)
Toothache	2 (12.5)	-	2 (4.3)
Backpain	-	2 (6.7)	2 (4.3)
Chills	1 (6.3)	1 (3.3)	2 (4.3)
Nosebleed	1 (6.3)	-	1 (2.2)
Cough	-	1 (3.3)	1 (2.2)
Anosmia	-	1 (3.3)	1 (2.2)
Shoulder pain	-	1 (3.3)	1 (2.2)
Tibia fracture	-	1 (3.3)	1 (2.2)
Cat scratching	1 (6.3)	-	1 (2.2)
Total	16 (100)	30 (100)	46 (100)

*n*—number of events; a RT PCR—reverse transcriptase polymerase chain reaction; SARS-CoV-2—severe acute respiratory syndrome coronavirus 2.

## Data Availability

The data presented in this study are available on request from the corresponding author.

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
