# Peer review of "Long-Term Immunogenicity and Safety of a Homologous Third Dose Booster Vaccination with TURKOVAC: Phase 2 Clinical Study Findings with 32-Week Post-Booster Follow-Up"

_vaccines, 2024, doi:10.3390/vaccines12020140_

Round 1
Reviewer 1 Report (Previous Reviewer 2)
Comments and Suggestions for Authors
Authors have well reponded to my comments and amended in the revision.
Reviewer 2 Report (Previous Reviewer 3)
Comments and Suggestions for Authors
I think the authors answered all Amy questions. I have no more questions/concerns on the manuscript.
Comments on the Quality of English LanguageMinor edit and proof reading are required for publication.
This manuscript is a resubmission of an earlier submission. The following is a list of the peer review reports and author responses from that submission.
Round 1
Reviewer 1 Report
Comments and Suggestions for Authors
The authors evaluated long-term immunogenicity and safety of a homologous booster vaccination with inactivated SARS-CoV-2 virus 43 participants. The authors measured NAbs and binding IgG antibody titers against WT virus. Comments for the authors below:
Major points:
1. Please include the information about the lineage prevalence during the trial.
2. Please describe the lineage infected to 9 participants and Nab titers before infection.
3. Please consider to include antibody titers against recent strains (XBB.1.16?).
Minor points:
1. P.2: Fist line of the Introduction: Coronavirus-19” should be “Coronavirus disease 2019”.
2. P.9: This sentence is incomplete. Please fix. “To our knowledge, this the first paper to report the outcomes in patients who were boosted with homologous”.
1. What is the main question addressed by the research?
The authors evaluated the booster dose of TURKOVAC in homologous immunization situation in phase 2 clinical study.
2. Do you consider the topic original or relevant in the field? Does it address a specific gap in the field?
This is an original topic and new but only tested against ancestral strain, which is not existing anymore.
3. What does it add to the subject area compared with other published material?
This manuscript contains new results, so worth publishing as a future refeence.
4. What specific improvements should the authors consider regarding the methodology? What further controls should be considered?
I suggest to measure antibodies against latest strain in Turkey.
5. Are the conclusions consistent with the evidence and arguments presented and do they address the main question posed?
As the authors only measured antibody against the ancestral strain, their evaluation of vaccine efficacy is not strong.
6. Are the references appropriate?
Yes.
7. Please include any additional comments on the tables and figures.
The graphs in Figure 1 are not clear.
Reviewer 2 Report
Comments and Suggestions for Authors
Authors studied the long-term efficacy and safety of a third TURKOVAC booster dose, given 12 weeks after the initial series. Antibody levels significantly increased from week 1, remaining elevated until at least week 4 (week 8 for whole SARS-CoV-2). Seroconversion persisted for 16 weeks for neutralizing antibodies and 32 weeks for SARS-CoV-2 IgGs. Nine participants (20.9%) tested positive between weeks 8 and 32, but none were hospitalized or died. This suggests TURKOVAC boosting provides effective protection against COVID-19 for at least 8 weeks, potentially reducing disease severity.
The authors referenced related articles titled 'Safety and immunogenicity of an inactivated whole virion SARS-CoV-2 vaccine, TURKOVAC, in healthy adults: Interim results from randomised, double-blind, placebo-controlled phase 1 and 2 trials. Vaccine. 2023;41(2):380-390.' and 'Long-Term Results of Immunogenicity of Booster Vaccination against SARS-CoV-2 (Hybrid COV-RAPEL TR Study) in Turkiye: A Double-Blind, Randomized, Controlled, Multicenter Phase 2 Clinical Study.' Authors are encouraged to compare their findings with these prior reports.
The quality of Figure 1 needs improvement, and Figure 2 is deemed unnecessary in this context. The novelty of the study may not be sufficient to support publication in this journal.
Comments on the Quality of English LanguageNo assay for the Quality of English Language.
Reviewer 3 Report
Comments and Suggestions for Authors
In this manuscript, Sezer et al., investigated the efficacy of heterologous boosting with TURKOVAC. Although there are some limitations including sample numbers and variants, they showed that TURKOVAC booster upregulated the neutralizing activity and IgG titer against SARS-CoV-2 WT. I have a few comments for this manuscript.
1. Figure 1: The resolution is low and it’s hard to read the word in the Figure. Please change it to the high-resolution one.
2. Figure 1: Why the values from pre-immune samples are under the threshold value (especially C and D)? Did these numbers still detectable?
3. Table 1: What the numbers shown in page 4 mean? And there is big space in the page 5. The numbers shown in page 4 should be in the page 5. Please clarify and fix this.
4. There are no Figure 2 legend description. Please add it.
5. Do the authors know if the participants had specific underlying diseases? If so, did the authors see any correlations the specific underlying diseases and immune response?
6. Which variants the participants were infected during sample correction? The authors described that “there were no hospitalizations or deaths due to COVID-19 throughout the 32-week study period despite the declining antibody GMTs over time”. Do the authors think this is the real effect of vaccine, not result of Omicron variants which shows mild symptoms?
Comments on the Quality of English LanguageModerate editing of English language required